# A qRT-PCR Method Capable of Quantifying Specific Microorganisms Compared to NGS-Based Metagenome Profiling Data

**DOI:** 10.3390/microorganisms10020324

**Published:** 2022-01-30

**Authors:** Jinuk Jeong, Seyoung Mun, Yunseok Oh, Chun-Sung Cho, Kyeongeui Yun, Yongju Ahn, Won-Hyong Chung, Mi Young Lim, Kyung Eun Lee, Tae Soon Hwang, Kyudong Han

**Affiliations:** 1Department of Bioconvergence Engineering, Dankook University, Yongin 16890, Korea; viem1273@gmail.com (J.J.); oys9807@gmail.com (Y.O.); 2Department of Nanobiomedical Science, Dankook University, Cheonan 31116, Korea; munseyoung@gmail.com; 3Center for Bio-Medical Engineering Core Facility, Dankook University, Cheonan 31116, Korea; 4Department of Neurosurgery, College of Medicine, Dankook University, Cheonan 31116, Korea; babyface@dankook.ac.kr; 5HuNBiome Co., Ltd., Seoul 08507, Korea; sindy@hunbiome.com (K.Y.); yongju.ahn@hunbiome.com (Y.A.); 6Research Group of Healthcare, Korea Food Research Institute, Wanju 55365, Korea; whchung@kfri.re.kr (W.-H.C.); mylim@kfri.re.kr (M.Y.L.); 7Department of Oral Medicine, School of Dentistry, Jeonbuk National University, Jeonju 54896, Korea; lke@jbnu.ac.kr; 8Theragen Bio Co., Ltd., Seongnam-si 13488, Korea; samuel.hwang@theragenbio.com; 9Department of Microbiology, College of Science & Technology, Dankook University, Cheonan 31116, Korea

**Keywords:** microbial diagnosis, next-generation sequencing, quantitative real-time PCR, metagenome

## Abstract

Metagenome profiling research using next-generation sequencing (NGS), a technique widely used to analyze the diversity and composition of microorganisms living in the human body, especially the gastrointestinal tract, has been actively conducted, and there is a growing interest in the quantitative and diagnostic technology for specific microorganisms. According to recent trends, quantitative real-time PCR (qRT-PCR) is still a considerable technique in detecting and quantifying bacteria associated with the human oral and nasal cavities, due to the analytical cost and time burden of NGS technology. Here, based on NGS metagenome profiling data produced by utilizing 100 gut microbiota samples, we conducted a comparative analysis for the identification and quantification of five bacterial genera (*Akkermansia*, *Bacteroides*, *Bifidobacterium*, *Phascolarctobacterium*, and *Roseburia*) within same metagenomic DNA samples through qRT-PCR assay in parallel. Genus-specific primers, targeting the particular gene of each genus for qRT-PCR assay, allowed a statistically consistent quantification pattern with the metagenome profiling data. Furthermore, results of bacterial identification through Sanger validation demonstrated the high genus-specificity of each primer set. Therefore, our study suggests that an approach to quantifying specific microorganisms by applying the qRT-PCR method can compensate for the concerns (potential issues) of NGS while also providing efficient benefits to various microbial industries.

## 1. Introduction

As a high-throughput nucleic acid sequencing technology, next-generation sequencing (NGS) has developed rapidly over the past 15 years. It is applied to various molecular genetics studies [1,2,3]. This technology is widely applied in molecular genetics and genomics fields to obtain genome sequences of various species and predict particular human diseases by identifying biomarker genes [4,5]. In addition, advances in NGS technology have recently brought new changes to microbiological research [6,7,8]. In past microbiological studies, microbial identification was possible through conventional Sanger sequencing using bacterial 16S rRNA gene or fungal ITS (internal transcribed spacer) regions [9]. However, conventional Sanger sequencing can only be used for culture-dependent microorganisms. On the other hand, NGS technology can be used to obtain genomes of multiple microbes (called metagenome), including culturable and unculturable microorganisms [10]. This approach for classifying microbiomes through metagenome sequencing technology (e.g., whole-metagenome sequencing and 16S rRNA sequencing) has been used to understand complex microbial communities and biological interactions [11,12]. Among several metagenome sequencing technologies, 16S rRNA gene sequencing is generally used because the 16S rRNA gene can be used as a phylogenetic marker to classify bacteria [13].

The 16S rRNA gene (approximately 1500 bp) consists of the nine hyper-variable regions (V1-V9) interspersed among highly conserved sequences. The use of these nine variable regions is a powerful tool with which to distinguish bacterial strains. Universal primer pairs, including a single or a combination of these regions (e.g., V1V2, V3V4, V4, and V5V6 regions on the 16S rRNA gene), can be used for 16S rRNA sequencing [14,15,16]. However, it is difficult to distinguish some bacterial species with high-sequence similarity based on sequences of partial 16S variable regions [17,18]. To overcome this bacterial misclassification problem, some companies (PacBio, Oxford Nanopore, and Loopseq) involved in NGS-based sequencing technology are developing novel metagenome analysis platforms that can read the full-length 16S rRNA gene to classify microorganisms [19,20,21]. There is continuous metagenome research and development to understand the association between microbiota (e.g., inhabiting in intestinal mucosa or oral cavity) and human health. Recently, human gut health has been investigated by profiling gut microbiome compositions using fecal samples [22,23,24].

Human gastrointestinal tracts are composed of more than 100 trillion different microorganisms such as bacteria, archaea, and viruses, creating a complex microbial ecosystem [25]. In the last ten years, large-scale gut microbiome studies such as the Human Microbiome Project (HMP) and MetaHIT have found that many human inflammatory diseases, obesity, and neurological diseases are associated with ‘dysbiosis’ [26,27,28]. The dysbiosis, which can be defined as imbalance of normal microbiota inhabiting the gut, it is mainly caused by bad eating habits, stress, and antibiotics. It can increase the proportion of intestinal harmful bacteria and yeast associated with various human diseases [29,30,31]. NGS-based microbial classification has been applied to various microbiome studies, resulting in significant biomedical findings. However, the metagenome sequencing technology has some issues. For example, it is expensive and time-consuming if the purpose is to detect specific microorganisms [32]. For these reasons, real-time PCR (also known as quantitative real-time PCR—qRT-PCR) is usually used to perform rapid microbial detection through target gene quantification from clinical samples taken from participants [33,34]. Indeed, many companies and researchers involved in microbial molecular diagnosis are attempting to overcome the issue of high-sequence similarity between bacterial species arising from 16S rRNA gene sequencing by developing specific primer pairs for use in qRT-PCR.

In this study, we investigated the concordance between qRT-PCR and NGS technology to confirm the efficiency of microbial quantification at the genus level associated with human health. Since NGS is currently unsuitable as a method for identifying species-level microorganisms, we confirmed the similarity of these two technologies at the genus level. Based on NGS metagenome profiling data from 100 gut microbiota samples, we conducted a comparative analysis of quantification accuracy for five bacterial genera (*Akkermansia*, *Bacteroides*, *Bifidobacterium*, *Phascolarctobacterium*, and *Roseburia*) through qRT-PCR assay in parallel. Genus-specific primer targeting of a particular gene (such as transcription termination/anti-termination protein; *nusG*) with one of the genera allows relatively similar quantification results with 16S V3-V4 metagenome profiling data. Additionally, we cross-validated the genus-specificity of the designed target primers via Sanger sequencing using qRT-PCR products as template DNAs.

## 2. Materials and Methods

### 2.1. Human Stool Samples Collection

One hundred human stool samples were collected using an OMNIgene∙GUT stool swab kit (DNAgenotek, Ottawa, Canada) from healthy adult men and women participating in the Korean Gut Microbiome Database Project of the Korea Food Research Institute (KFRI, Wanju, Korea). All stool samples were then stored at −80 °C for the experimental downstream processing. The sampling was carried out with the prior informed consent of all participants before this study began. None of participants affected the research results, such as taking medication before the study began. The clinical sample collection in this study (stool samples collection) was approved by the ethics committee of Theragen Bio (Theragen Bio, Seongnam, Korea) Institutional Review Board (IRB Protocol Number: 700062-20180905-JR-005-01). All clinical experiments applied to this study were carried out according to the guidelines and regulations of the declaration of Helsinki.

### 2.2. Metagenomic DNA Extraction

Total metagenomic DNA (mDNA) from 100 stool samples was isolated using a QIAamp DNA Stool Mini Kit (Qiagen, Hilden, Germany), and all experimental processes were performed in accordance with the optimal protocols provided on the DNA extraction kit. The quality check of all isolated mDNA was conducted using a Bioanalyzer (Agilent 2100, Santa Clara, CA, USA) at the Center for Bio-medical Engineering Core Facility (Dankook University, Yongin, Korea). All mDNA samples were then stored at 4 °C until the following process.

### 2.3. Illumina 16S V3–V4 Amplicon Sequencing Library Preparation and Sequencing

A total of 100 metagenome sequencing libraries were prepared according to the Illumina 16S amplicon sequencing library construction workflow (Illumina, San Diego, CA, USA). The Illumina platform targeted an area containing the V3-V4 hyper-variable region of the bacterial 16S rRNA gene. The PCR amplification of the target region was started immediately after the mDNA was extracted. The 16S V3-V4 amplicon was amplified using KAPA HiFi Hot Start Ready Mix (2×) (Roche, Penzberg, Germany). For this purpose, a pair of V3-V4 target-specific universal primers recommended by Illumina were used. The primer sequences were as follows:

16S 341F forward primer is 5′-TCGTCGGCAGCGTCAGATGTGTATAAGAGACAGCCTACGGGNGGCWGCAG-3′ and 16S 806R reverse primer is 5′-GTCTCGTGGGCTCGGAGATGTGTATAAGAGACAGGACTACHVGGGTATCTAATCC-3′. After the PCR amplification, the clean-up process of all PCR products was conducted using the AMPure XP beads (Beckman Coulter, California, USA). Then, additional PCR amplification was performed to add multiplexing indices and Illumina sequencing adapters using the Nextera XT Index Kit (Illumina, San Diego, CA, USA). The final PCR products were then purified once again using AMPure XP beads. After the amplicon library construction, the 16S metagenome sequencing was carried out using the paired-end 2 × 300 bp Illumina MiSeq protocol (Illumina MiSeq, San Diego, CA, USA; [35]). Finally, all Illumina sequencing raw data for the 16S V3-V4 metagenome sequencing were deposited in the NCBI Sequence Read Archive database (biosample accession number: “PRJNA744351”; SRR number: 146 SRR15067184 to SRR15067283).

### 2.4. Bacterial Genus-Specific Primer Design Methods

Genus-specific primers for qRT-PCR amplification were designed to identify and absolutely quantify the particular five bacterial genera from mDNA. The overall process for primer design in this study is as follows.

#### 2.4.1. Bacterial Genera and Target Gene Selection

Based on 16S V3-V4 metagenome sequencing profiling data at the genus level, the *Akkermansia*, *Bacteroides*, *Bifidobacterium*, *Phascolarctobacterium*, and *Roseburia* were selected for this study. These five bacterial genera are present in the human gut microbiota and are probiotic bacterial strains closely related to obesity, dysbiosis, and inflammatory bowel disease prevention [36,37,38,39,40]. They are also beneficial bacteria that contribute significantly to the production of short-chain fatty acids (SCFA; e.g., acetic, butyric, and propionic acid), which are key to human health control, such as increasing immunity, maintaining intestinal homeostasis, and preventing fat accumulation in the body [41]. Considering these interesting bacterial features, we studied these five selected genera to compare the relative proportions of particular bacterial load within the samples, measured using both the qRT-PCR assay and NGS frequency data. Then, we selected particular genes (*nusG*, *transaldolase*, and *ddl*) to detect each bacterial genus through qRT-PCR.

#### 2.4.2. Sorting of NCBI Annotation Information

The National Center for Biotechnology Information Database (NCBI DB) was used to obtain sequence information about the selected genes to target each bacterial genus. First, the ‘Identical Protein Groups’ category was used to extract a comprehensive summary table that explains the gene annotation information on the NCBI DB (e.g., nucleotide and protein accession number, organism at species and strain level, etc.). Next, the ambiguously classified information on the table, such as ‘hypothetical protein’, was excluded from the list. Then, repetitive information about the randomly selected bacterial species (included at the strain level) belonging to each genus was filtered. Finally, all ‘Protein IDs’ on the filtered list were isolated to obtain the amino acid sequence information needed to convert to gene sequence.

#### 2.4.3. Extraction of Coding Sequence Information

The Batch Entrez open web bioinformatics tool linked with NCBI DB (www.ncbi.nlm.nih.gov/sites/batchentrez, accessed on 26 April 2020) was used to convert amino acid information about specific genes into coding sequences (CDS) in the form of the FASTA format.

#### 2.4.4. Multiple Sequence Alignment and Selection of Target-Specific Regions

To find the consistent sequence regions for qRT-PCR primer design, the multiple sequence alignment of the CDS information was performed using the BioEdit 7.2v software.

#### 2.4.5. In Silico Test

The experimental suitability (Tm value, GC%, and potential for primer--dimer to form, etc.) for qRT-PCR of the selected primer pairs were checked using the Oligo calc and Oligo Analysis open web tool (http://biotools.nubic.northwestern.edu/, accessed on 26 April 2020; http://www.operon.com/tools/oligo-analysis-tool.aspx, accessed on 26 April 2020). Next, the NCBI Nucleotide BLAST (https://blast.ncbi.nlm.nih.gov, accessed on 27 April 2020) was used to confirm the primer specificity for the targeted bacterial genus.

### 2.5. Bacterial Quantification Using qRT-PCR

The qRT-PCR was conducted to quantify the five bacterial genera frequencies from the isolated 100 mDNA samples. First of all, double-stranded DNA concentration within all mDNA samples was measured using the Qbit Fluorometer 4.0 v at the Center for Bio-medical Engineering Core Facility (Dankook University, Yongin, Korea) and 1X dsDNA HS Assay kit (ThermoFisher Scientific, Waltham, MA, USA). Then, all template mDNA was then normalized to an identical concentration (10 ng/μL) via dilution by using distilled water. Next, a standard curve for 100 diluted DNA samples was calculated to confirm that the template DNA concentration used in qRT-PCR was consistently normalized. In this step, the cycle threshold (Ct) value for amplicon quantity in qRT-PCR measured using the bacterial 16S V4 primer pair (515F and 806R) was applied. These Ct values were reflected from the 10-fold serial dilution (10-1, 10-2, and 10-3 dilution) samples for each formerly normalized mDNA sample. Finally, the bacterial genera frequency within 100 template mDNA was confirmed using the StepOnePlus™ Real-Time PCR (ThermoFisher Scientific, Waltham, MA, USA) at the Center for Bio-medical Engineering Core Facility (Dankook University, Yongin, Korea) and QuantiSpeed SYBR No-Rox kit (PhileKorea, Seoul, Korea). Each qRT-PCR primer annealing temperature condition applied at this step is as follows; *Akkermansia*: 65 °C, *Bacteroides*: 65 °C, *Bifidobacterium*: 62.5 °C, *Phascolarctobacterium*: 62 °C, and *Roseburia*: 65 °C. Each summary table (Appendix A), which included the Ct value of five bacterial genera, was used to compare NGS-based bacterial frequency data.

### 2.6. Sanger Sequencing

The Sanger sequencing was conducted to verify the molecular specificity and experimental accuracy of the five primer pairs designed for this study. First of all, the ten qRT-PCR amplicon samples (Top five and bottom five of the measured Ct value) were sorted from each qRT-PCR summary table (Appendix A) for the Sanger validation. The selected qRT-PCR amplicon samples were then purified using the FavorPrep™ GEL/PCR Purification Kit (Favorgen, Tiwan). Next, molecular cloning, through ligation, and a transformation process were carried out using TOPcloner™ TA Kit (Enzynomics, Daejeon, Korea) and DH5α chemically competent *E. coli* (Enzynomics, Daejeon, Korea) to obtain templates DNA containing amplicon sequences. A colony PCR amplification using the M13 primer pair (M13F: 5′-GTAAAACGACGGCCAG-3′; M13R: 5′-CAGGAAACAGCTATGAC-3′) was conducted to check whether the target sequence was inserted in the plasmid cloning vector. We applied the TOPO TA vector with M13 primer regions (M13F and M13R; approximately 200 bp) because the input library length for high-quality Sanger sequencing results is at least 400 bp [42]. The colony PCR products were once again purified through the FavorPrep™ GEL/PCR Purification Kit (Favorgen, Tiwan). The Sanger sequencing with purified products as template DNA was carried out using the ABI 3500XL Genetic Analyzer sequencer (ThermoFisher Scientific, Waltham, MA, USA) at the Center for Bio-medical Engineering Core Facility (Dankook University, Yongin, Korea). The ABI and FASTA format data generated after sequencing were used as input data in the NCBI Nucleotide BLAST tool for bacterial genus identification.

### 2.7. 16S V3–V4 Data Processing and Microbial Community Analysis

The 16S V3-V4 sequencing reads were demultiplexed using the split_libraries_fastq.py function in QIIME2 metagenome analysis pipeline and sequences were quality trimmed using the Divisive Amplicon Denoising Algorithm 2 (DADA2) pipeline in R (version 3.3.2) with the parameters; EE = 2, TruncL = c (200, 180) and q = 10. The set of unique 16S V3-V4 DNA sequences, referred to as amplicon sequence variants (ASVs), were then inferred using DADA2 and an ASV table of read counts per ASV per sample was generated. ASV taxonomies were taxonomically classified using the sklearn-based naive Bayes classifier with the SILVA 138v 16S rRNA database.

### 2.8. Statistical Analysis

An association analysis of the bacterial genera proportions data within 100 samples, measured from the qRT-PCR assay and 16S metagenome sequencing results, was conducted through a statistical test using the Spearman correlation test. The Spearman correlation’s statistical significance evaluated the statistical similarity between the two methods was denoted as asterisk (*) if *p*-value < 0.05 and R value (Spearman’s value) > 0.5.

## 3. Results

### 3.1. Selection of Five Bacterial Genera from 16S Metagenome Analysis Data

This study confirmed the consistency of the relative proportion pattern within samples of five particular bacterial genera between 16S metagenome profiling data and qRT-PCR quantitative results (Figure 1). We collected stool specimens from 100 healthy Korean (adult men and women). We then extracted mDNAs from these 100 specimens and simultaneously checked DNA quality (e.g., DNA degradation, concentration, and purity) for accurate microbial quantification. To select bacterial genera for quantifying their frequencies in the specimens, we successfully prepared the Illumina 16S V3-V4 short-read amplicon sequencing libraries using these 100 mDNA samples. NGS-based 16S metagenome sequencing was performed.

As a result of 16S metagenome sequencing, the average number of demultiplexed reads generated from 100 samples was 133,288, of which 65,892 were filtered through the DADA2 pipeline with non-chimeric reads (Appendix A). After 16S V3–V4 sequencing data processing, the total number of bacterial ASV taxonomy classified from the SILVA 138v 16S rRNA gene reference database was 6902 (with more 70% classification confidence threshold), of which 338 were classified at the genus level (Appendix A). Five bacterial genera (*Akkermansia*, *Bacteroides*, *Bifidobacterium*, *Phascolarctobacterium*, and *Roseburia*) were then selected based on 16S metagenome profiling data obtained previously (described in detail in Section 2).

### 3.2. Bacterial Genus-Specific Primer Design

To quantify the bacterial load in each sample using qRT-PCR assay, we designed genus-specific primer pairs to detect those five bacterial genera (Table 1). Considering that high-sequence similarity of the 16S rRNA gene internal hyper-variable region would make it difficult to detect a specific genus, we used sequence information of certain bacterial gene domains annotated on the NCBI reference database (Appendix A; details shown in Section 2).

First, we applied sequence information of the *nusG* gene encoding a transcription termination/antitermination protein essential for bacterial transcription [43]. For the *Bifidobacterium* and *Akkermansia*, it was challenging to find consistent regions from multiple sequence alignment results between bacterial species belonging to each genus. Therefore, we reselected the transaldolase and D-alanine D-alanine ligase (*ddl*) genes as target-specific genes for *Bifidobacterium* and *Akkermansia*, respectively. In the case of *Bifidobacterium*, we referred to previous studies showing that *Bifidobacterium* species had a gene-coding region capable of expressing at least 14 types of transaldolase that could be differentiated using protein electrophoresis [44]. Additionally, we selected the *ddl* gene as a target gene for *Akkermansia*, considering that it is an essential factor for bacterial cell wall synthesis [45]. Finally, we conducted an in silico test using the NCBI Nucleotide BLAST web tool to validate the bacterial genus-specificities of five primer pairs designed for this study (Appendix A). Although in silico test results showed classification results for microorganisms other than the targeted bacterial genus (e.g., *Raphanus sativus*, *Rodentibacter pneumotropicus*, and *Acetobacterium woodii*), we determined that they would not affect our qRT-PCR assay because they were eukaryotic organisms, not human gut bacteria [46,47,48].

### 3.3. Quantification and Normalization of Metagenomic DNA

Before estimating a particular bacterial load within each human fecal sample, we normalized bacterial DNA concentration, which was included in the 100 mDNA samples, through a standard curve calculation using the qRT-PCR (Figure 2; Table 2; Appendix A). We used 10-fold serially diluted mDNA samples (10^−1^, 10^−2^, and 10^−3^ dilution from 10 ng/μL concentration of dsDNA per each sample) and bacterial 16S V4 universal primers (515F and 806R) for this qRT-PCR assay. Theoretically, considering that the 16S rRNA gene is the most conserved region in the bacterial genome, it would be appropriate to use the 16S V4 specific-primer pair for internal bacterial genomic DNA quantification of mDNA samples [49,50]. Results of the qRT-PCR assay confirmed that average Ct values for V4 amplification using 10^−1^, 10^−2^, and 10^−3^ diluted samples were measured to be 27.31, 23.26, and 19.85, respectively. In addition, we confirmed that the standard deviation (SD) and the coefficient of variation (CV) value of each Ct value for those three dilution factors were 0.55 and 2.41%, respectively.

Calculating the standard curve based on these measurements, the R^2^ value of the trend line connecting Ct values for all 10-fold diluted samples was about 0.97. Additionally, we found that the R^2^ value of the trend line was close to 1 when calculating the standard curve based on the average Ct value. These results showed that our systematic quantification challenge for the particular genera was conducted with the quantitatively normalized bacterial DNA samples.

### 3.4. Parallel Comparison of qRT-PCR and 16S Metagenome Profiling Data

We compared the relative bacterial proportion similarity of the five genera within the human gut microbial community between the qRT-PCR assay using formerly normalized 100 mDNA samples (10 ng/μL per sample) and the NGS-based 16S metagenome profiling data (Figure 3 and Figure 4, Table 3, Appendix A). In order to compare both approaches, we distributed quantitative figures using a formula of 1/2^Ct to measure the potential bacterial frequency of genus within each sample based on the Ct values exported by qRT-PCR analysis (Shown in Appendix A; [51]).

Comparing results of qRT-PCR assay with NGS proportions data in parallel, we found that the relative proportion patterns of five bacterial genera were remarkably consistent between the two different quantification methods. We highlight that four bacterial (*Akkermansia*, *Bifidobacterium*, *Phascolarctobacterium*, and *Roseburia*) genera present in low proportions in 100 samples of human gut microbiota also showed significant relative proportions in quantification results. Additionally, we verified an association between bacterial proportions data measured with two different quantification methods using the Spearman correlation statistical analysis to supplement these parallel comparison results.

Results of statistical analysis confirmed that the Spearman correlation values for all bacterial genera groups were significantly calculated (*p*-value < 0.05; Spearman’s rho value > 0.5). Considering that the Spearman analysis evaluates statistically positive similarity if the correction value (Spearman’s rho value) is 0.4 or higher, we could confirm that the relative bacterial proportion measured with NGS is similar to that measured with qRT-PCR assay [52]. Our comparative results showed that the five genus-specific primer pairs had high-binding sensitivity and specificity for the target mDNA.

### 3.5. Verification of Primer Specificity

We performed Sanger sequencing for qRT-PCR amplicon products to confirm bacterial genus-specificity of five primer pairs designed for this study. Sequencing results were verified by BLAST test (Basic Local Alignment Search Tool; Appendix A; Table 4; Appendix A).

When the Sanger sequencing results were confirmed through the BLAST search using the blastn parameter, we found that the taxonomy definition rate with the NCBI DB was almost 100% in the four groups (high and low proportion groups of the *Akkermansia*, *Bacteroides*, *Phascolarctobacterium*, and *Roseburia*). For *Bifidobacterium*, the bacterial taxonomy definition rates of high and low groups were 92%, and 60%, respectively. Although some BLAST results of *Bifidobacterium*_low groups were undefined with NCBI DB, we found that the relative proportion in 100 people calculated from the NGS frequency was almost similar to that based on qRT-PCR Ct value. As a result of Sanger validation, qRT-PCR and NGS proportion data shown in the Appendix A of the *Bifidobacterium*_low group samples for which the NCBI nucleotide BLAST search was not valid (less than 60% of bacterial taxonomy definition rate per each plate) were as follows: Bif_Mi_01 (Candidate 50), 0.01 and 1.80; Bif_Mi_02 (Candidate 77), 0.01 and 0.00; Bif_Mi_03 (Candidate 83), 0.01 and 0.00. In this regard, this issue found in the *Bifidobacterium*_low group was likely to result in low transformation efficiency in the TA cloning process due to the following: (i) too-low bacterial frequency within the samples; and (ii) high G + C content (approximately 63 mol%) in the transaldolase gene coding region of the *Bifidobacterium* genome rather than misbinding of the specific primer pairs that occurred during qRT-PCR amplification [53,54,55]. Considering that identification of the target bacterial genus was possible in all Sanger sequencing validation results of the selected genus groups reflecting high and low genus proportions within each sample, we confirmed that the five specific primer pairs designed for this study were sufficient for bacterial detection and quantification using qRT-PCR assay.

## 4. Discussion

Since the Human Genome Project, high-throughput sequencing-based microbiome research projects such as the HMP and the MetaHIT consortium have emphasized the importance of identifying the association between microorganisms and human diseases. In particular, these projects explained that causes of various physical diseases are closely related to the “dysbiosis” phenomenon, in which normal intestinal microbiota is unbalanced [26,56]. Accordingly, various probiotic therapy-associated industries and researchers have classified human gut microbial compositions using NGS technology to identify relative intestinal proportions of beneficial and harmful bacteria and their biological effects [57,58]. However, NGS is expensive and time-consuming. These are considered important issues in microbiome-associated health care industries. Furthermore, microbial misclassification problems at the species level, caused by high sequence similarity within the 16S rRNA gene, is also a challenge to be addressed [18,59]. Therefore, some researchers have suggested that running experiments in parallel with qRT-PCR, which enables accurate and rapid detection and quantification of the target genes, is a solution to compensate for these issues of the NGS technology. Due to the high fluorescence sensitivity of the real-time PCR machine enabling accurate detection of specific microbial species in the sample, the qRT-PCR assay is considered the ‘gold standard method’, especially in the molecular microbial diagnostics and probiotics fields [60,61].

Through parallel comparison analysis, it was confirmed that the qRT-PCR technique could be an excellent alternative to existing NGS-based 16S V3-V4 metagenome sequencing methods, as it could enable reliable detection and quantification (for the gut-associated microbiota: *Akkermansia*, *Bacteroides*, *Bifidobacterium*, *Phascolarctobacterium*, and *Roseburia*). Since quantifying gut microbiome compositions of healthy subjects is necessary to establish a baseline against which microbiome changes could be detected under pathological conditions, we performed normalization to the initial microbial density using the V4 hyper-variable region on the 16S rRNA gene, and thus minimized quantification errors in specific bacterial populations through this process by quantifying the total load of the 16S rRNA genes in the meta-samples [50]. The qRT-PCR quantification process using the internal standard method in bacterial quantitation systems could avoid high-rate amplification bias in NGS-based metagenome sequencing and is essential to increase the reproducibility thereof [51]. Although the 16S rRNA gene primer is specifically designed and optimized for qRT-PCR or NGS, it is recommended to consider potential bias and the miss annealing due to sequence similarity between specific bacterial strains for qRT-PCR. In our study, the qRT-PCR primers targeted CDS sites of a gene (e.g., bacterial housekeeping genes such as *nusG* or *ddl*) representing a particular microbial genus to ensure high sensitivity and reduce biased quantitative results due to sequence similarity. In addition, we selected five amplicon samples with the high- and low-frequencies generated after qRT-PCR amplification, respectively, and conducted Sanger validation to evaluate the accuracy and sensitivity of the designed genus-specific primers. As a result, almost all amplicon samples correctly matched the target bacterial taxonomy at the genus level, suggesting the specificity and suitability of our target primers for particular genus quantification. We assumed that several undefined samples through Sanger sequencing validation might not be identified due to the reduced PCR cloning efficiency caused by the high G + C contents of the target CDS or the minimal DNA content in *Bifidobacterium*. Nonetheless, the results of evaluating the specificity and sensitivity of primers used to quantify microorganisms using Sanger sequencing validation suggest that this validation process is an essential task. As a standard research procedure for clinical diagnostic research and industrial development. Results of cross-checking the correlation of measured bacterial proportions between two different quantification methods through Spearman correlation statistical analysis also verified that the qRT-PCR method was sufficient and accurate for quantitative analysis of specific microbial genera in meta-samples. Therefore, we propose an efficient system in various fields requiring rapid quantification of indicator microorganisms, such as medicine, agriculture, and marine biology, etc.

In conclusion, through a parallel comparative analysis with 16S metagenome sequencing data, we confirmed that the qRT-PCR method could overcome some issues (such as the high analytic cost burden or time consumption) of the NGS technology applied in various microbial industries. Our findings showed that relative bacterial proportions of five genera within samples measured via qRT-PCR performed under normalized mDNA concentration conditions were statistically similar to those based on 16S metagenome sequencing data. Additionally, we cross-validated bacterial genus-specificity of the five primer pairs designed for the qRT-PCR assay through Sanger sequencing and NCBI BLAST test for bacterial identification. In this respect, we claim that the qRT-PCR method is one of appropriate tools for identifying the relative proportions of particular microorganisms within a sample. Furthermore, our results suggest that applying qRT-PCR to specific microbial validation in the NGS-based microbial diagnosis industry can compensate for concerns of NGS technology by providing economical, fast, and accurate services to consumers in terms of turnaround time.

## Figures and Tables

**Figure 1 microorganisms-10-00324-f001:**
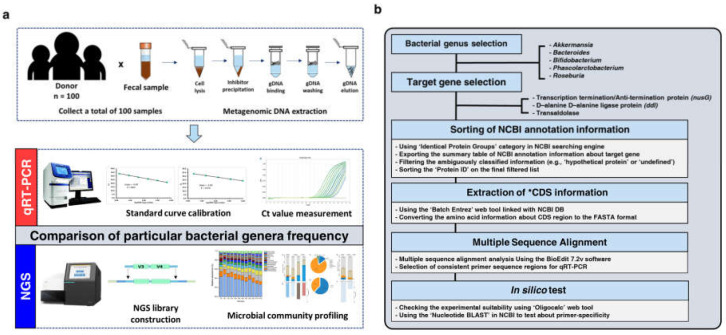
Experimental introduction in this study and schematic workflow of a genus-specific primer design method for qRT-PCR assay. Using stool samples obtained from 100 healthy adults, we performed a comparative analysis to evaluate the accuracy of microbial quantification by two different molecular technologies. (**a**) The overall experimental workflow of the qRT-PCR assay and NGS-based 16S V3-V4 metagenome sequencing. (**b**) Schematic diagram showing the process to design the genus-specific primer set to quantify the proportions of the five selected bacterial genera within each sample using qRT-PCR.

**Figure 2 microorganisms-10-00324-f002:**
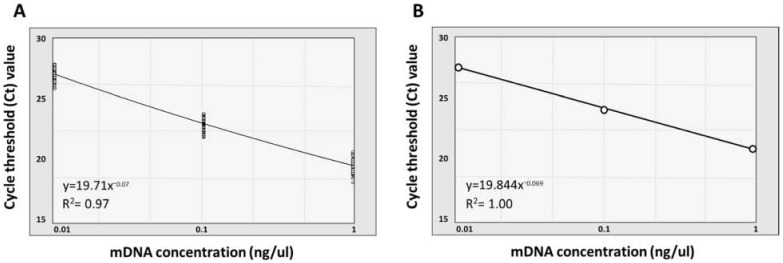
Standard curve calculation to confirm normalization of the mDNA concentration used for qRT-PCR analysis. Each double-stranded mDNA sample diluted with a three step 10-fold serial dilution (10^−1^, 10^−2^, and 10^−3^) was standardized at 10 ng/μL for standard curve calculation. The graph’s *x*-axis indicates the concentration of each 10-fold diluted mDNA used in standard curve calculation, and the *y*-axis shows the Ct value measured from the qRT-PCR. (**A**) Standard curve graph representing the Ct value of every sample with 10-fold serially diluted mDNA concentration (R^2^ = 0.97). (**B**) Standard curve graph representing the average Ct value of each concentration of the diluted mDNA sample (R^2^ = 1.00).

**Figure 3 microorganisms-10-00324-f003:**
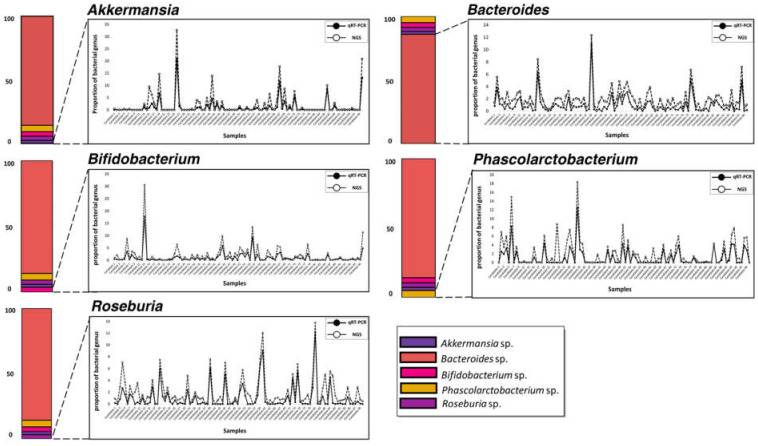
Parallel comparison of the five bacterial genera proportions measured from two different quantification methods. Multiple overlaid connected line graphs show the quantitative identity of each bacterial genus measured by both methods. The graph’s *x*-axis indicates the 100 samples (denote as ‘candidate’), and the *y*-axis indicates the relative proportion value for each particular bacterial genus. The bar plot shown on the left side represents an average abundance of the five bacterial genera calculated from 16S metagenomic profiling analysis.

**Figure 4 microorganisms-10-00324-f004:**
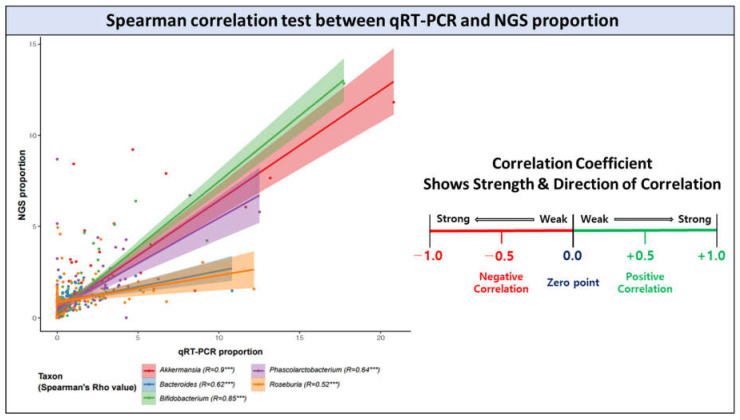
Spearman correlation scatter plot showing the relationship of each bacterial relative proportion value measured by qRT-PCR assay and 16S V3-V4 metagenome sequencing data; *** = Spearman *p*-value < 0.001, R = Spearman’s rho value.

**Table 1 microorganisms-10-00324-t001:** Overall information of bacterial genus-specific primer set.

Bacterial Taxon	Rank	Target Gene	Foward Primer (5′-3′)	Reverse Primer (5′-3′)	Tm °C (F/R)	GC % (F/R)	Amplicon Size (bp)
*Akkermansia*	genus	*ddl*	CTTCGTGCTGGAAATCAACACC	CGATAATTCCGCTATTTTTTCGC	62.1/59.2	50/39	135
*Bacteroides*	genus	*nusG*	GGTGCCTCTCAGACAATCAG	CAATGATACCACTGAATCCGCT	60.5/60.1	55/45	149
*Bifidobacterium*	genus	Transaldolase	AAGGGCATCTCCGTCAACG	GGAGACGAAGAAGGAAGCGA	59.5/60.5	58/55	146
*Phascolarctobacterium*	genus	*nusG*	TTCCTGGTTATGTGCTTGTAGAG	CAGTCAAAGGAATCGGTTTAGTA	60.9/59.2	43/39	114
*Roseburia*	genus	*nusG*	AAATACCCGTGGTGTTACCG	GTGTCTCCCTCTGTAAAGTCA	58.4/59.5	50/48	130

**Table 2 microorganisms-10-00324-t002:** Average standard curve calculation results using qRT-PCR assay.

Dilution Factor	Average * Ct Value	* SD Value	* CV Value	Target Gene
10^−3^ from 10 ng	27.31	0.49	1.78	16S rRNA V4 region
10^−2^ from 10 ng	23.26	0.52	2.24	16S rRNA V4 region
10^−1^ from 10 ng	19.85	0.64	3.21	16S rRNA V4 region

* Ct value—cycle threshold value; * SD value—standard deviation value; * CV value—coefficient of variation value.

**Table 3 microorganisms-10-00324-t003:** Statistical result of the Spearman correlation test between two different quantification methods.

	Spearman Correlation Test		
Bacterial Genus	* R Value	Spearman *p*-Value	* Spearman’s Sig.
*Akkermansia*	0.895622663	2.98 × 10^−36^	***
*Bacteroides*	0.624122412	0	***
*Bifidobacterium*	0.853890597	1.51 × 10^−29^	***
*Phascolarctobacterium*	0.644456804	4.67 × 10^−13^	***
*Roseburia*	0.518642542	3.25 × 10^−8^	***

* Sig.: Statistical significance; asterisk (***) if *p*-value < 0.0001; * R: Spearman’s rho value.

**Table 4 microorganisms-10-00324-t004:** Bacterial identification result by Sanger sequencing.

Bacterial Genus	Defined Bacterial Taxon Counts in NCBI Database			Defined Bacterial Taxon Rates (%) of Sanger Validation		
	High Top 5 (Ct Value)	Low Top 5 (Ct Value)	Total	High Top 5 (Ct Value)	Low Top 5 (Ct Value)	Total
*Akkermansia*	25	25	50	100.00	100.00	100.00
*Bacteroides*	25	24	49	100.00	96.00	98.00
*Bifidobacterium*	23	15	38	92.00	60.00	76.00
*Phascolarctobacterium*	25	25	50	100.00	100.00	100.00
*Roseburiea*	25	25	50	100.00	100.00	100.00

## Data Availability

The information of metagenome datasets generated during the current study are included in this article and it is available from the corresponding author on reasonable request.

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
