# Peer review of "A qRT-PCR Method Capable of Quantifying Specific Microorganisms Compared to NGS-Based Metagenome Profiling Data"

_microorganisms, 2022, doi:10.3390/microorganisms10020324_

Round 1
Reviewer 1 Report
The work entitled “Evaluation of an effective detection and quantification method for particular microorganisms by comparing NGS-based metagenome profiling data” by Jeong et al., describes the comparison between “qRT-PCR and next-generation sequencing (NGS) technology to confirm the efficiency of microbial quantification for genus level associated with the human health.” Authors select five bacterial genera for this purpose (Akkermansia, Bacteroides, Bifidobacterium, Phascolarctobacterium, and Roseburia) present in 100 gut microbiota samples (stool samples).
The load of this work is essentially methodological. Authors describe in detail all the experimental steps needed to carry out this comparative study. This is correct but makes the manuscript highly involved and I guess unfriendly for some readers of this journal. Although I am positive for the publication of this work, my recommendation is that the text should be made more accessible to other different, potential readers. Authors should make an extra effort and rewrite the manuscript trying to reach those other potential readers.
Many comments are present in the attached pdf. Authors should address all of them. To begin with, the very title is difficult to interpret. Authors should choose a direct and easily interpretable title.
Figures are correct but should have enough resolution to make visible small characters (mainly Figure 1). Importantly: authors should pay attention to the format of the journal, in particular, references:
- Author 1, A.B.; Author 2, C.D. Title of the article. Abbreviated Journal Name Year, Volume, page range.
Regarding to this: authors should try to properly select the references. Just as an example: references 22 and 23 are selected for supporting the claim “… association between microbiota (e.g., inhabiting in intestinal mucosa or oral cavity) and human health… “ . There are plenty of excellent reviews concerning this particular point. This can also apply for references 24-26.

Reviewer 2 Report
The manuscript entitled “Evaluation of an effective detection and quantification method for particular microorganisms by comparing NGS-based meta- genome profiling data” intends to show the relevance of performing a dual approach (qPCR and NGS) to analyse the human gut microbial communities, in order to make a more accurate assessment of gut health. The authors present the results of NGS of 100 human faecal samples, and from these results they selected 5 bacterial genera to analyse by qPCR. They designed specific primers for each of the 5 bacteria/selected genes, and successively showed their specificity. However, I have some concerns regarding the manuscript before it can be published.
Lines 253-263: I think that should be better explained in the methods rather than in the results. Same for lines 361-369.
Lines 326- 327: I don´t understand the analysis of qPCR data that the authors have performed. Why have the authors done the analysis “we calculated the potential bacterial frequency (1/2Ct value)”? This is quite basic and I don´t get the point. Why have the authors not calculated the amount of target gene in each sample by using a series of dilutions from plasmid DNA containing a fragment of your gene of interest? (see for ex. https://www.ncbi.nlm.nih.gov/pmc/articles/PMC6549245/).
Figure 5. This figure represents a very basic knowledge of molecular biology (cloning of a fragment of interest). I don´t think this figure is necessary.
The language is not of enough quality and should be revised.
Some examples:
line 23: “… real-time PCR (qRT-PCR) are still of considerable technique in detecting…”. Should be something like: “… real-time PCR (qRT-PCR) is still an important technique for detection and…”
Line 60: “However, there is an issue that reading partial 16S variable…” It must be corrected.
Round 2
Reviewer 1 Report
In this new version of the manuscript, authors have addressed the points I raised previously. Now, I have included some minor corrections that authors should check (attached pdf). In general, the manuscript is more readable and accessible to the readers of the journal which are not familiar with the described experimental approaches.
My recommendation is that after those minor corrections are included, the work is acceptable for publication.

Reviewer 2 Report
The authors have extensively edited the manuscript, and the language improved considerably. I think that the manuscript is much more clear now, and I recommend its publication.
